

# Emergence of visible light optical properties of L-phenylalanine aggregates

Mantas Ziaunys and Vytautas Smirnovas

Vilnius University, Life Sciences Center, Institute of Biotechnology, Vilnius, Lithuania

## ABSTRACT

The ability of phenylalanine to form fibrillar nanostructures was demonstrated on multiple occasions, and such an oligomerization reaction could be the cause of cytotoxicity in patients with phenylketonuria. These findings were supported by claims that L-phenylalanine (Phe) fibrils have amyloid properties and can be detected using thioflavin T fluorescence assay. However, a part of Phe aggregation studies reported the opposite data, suggesting no amyloid structures to be formed. Due to the contradicting reports, the amyloid nature of Phe aggregates remains uncertain. In this work we tested Phe aggregation under conditions where amyloid formation was previously reported. We show the emergence of Phe aggregates with visible light optical properties that overlap with the spectra of dyes used in amyloid fibril assays, which could lead to false-positive identifications.

## INTRODUCTION

Amyloid fibrils are linked to several neurodegenerative disorders, such as Alzheimer's or Parkinson's disease (*Chiti & Dobson, 2006*), and their formation is the result of protein aggregation into highly ordered structures (*Fitzpatrick et al., 2013*). It was shown that full-length amyloid proteins are not required for such aggregation to occur (*Halverson et al., 1990*; *von Bergen et al., 2000*). In order to find the minimum possible peptide capable of fibril formation, shorter and shorter fragments were tested, all the way down to dipeptides (*Azriel & Gazit, 2001*; *Reches & Gazit, 2003*; *Gazit, 2007*). Eventually it was demonstrated by *Adler-Abramovich et al. (2012)* that even unmodified monomeric L-phenylalanine (Phe) is capable of forming fibrillar aggregates, which share similarities to amyloids. The amyloidogenic nature of Phe fibrils were supported by experimental data, showing the presence of beta-sheet conformation (by CD) (*Smith et al., 2008*), and other amyloid-specific properties, such as thioflavin T (ThT) and Congo Red binding ability and even seeding potential (*Singh et al., 2014*; *Shaham-Niv et al., 2015*; *Anand et al., 2017*). Different models of aggregate assembly and structure types were also presented (*Mossou et al., 2014*; *German, Uyaver & Hansmann, 2015*; *Do, Kincannon & Bowers, 2015*). However, later *Perween et al., (2013)* suggested that the CD spectra may be interpreted by carboxyl group (n–π*) and benzene ring (π–π*) transitions and was not the result of amyloid assemblies. It was also shown that these fibrils only form at the edges of confocal microscopy samples due to crystallization (*Perween et al., 2013*) and that they

Corresponding author
Vytautas Smirnovas,
vytautas.smirnovas@bti.vu.lt

do not exist in solution (*Griffith et al., 2015*). The conditions used to generate these fibrils were varied, with some using low concentrations of Phe and others creating supersaturated hydrogels (*Adler-Abramovich et al., 2012*; *Perween et al., 2013*; *Singh et al., 2014*; *Shaham-Niv et al., 2015*; *De Luigi et al., 2015*; *Banik et al., 2016*, *2017*).

Thioflavin T fluorescence was the major assay to confirm Phe amyloid fibril formation, however, the possibility of a false-positive detection was not considered. Hydrogel formation could immobilize ThT and increase its fluorescence potential (*Hutter et al., 2011*) in a similar way as binding to amyloids would, so it should be considered in case of aggregation studies at high Phe concentrations. Autofluorescence of Phe aggregates is another factor to be considered. Previously reported Phe aggregation models display π-stacking as a large contributor to stable aggregate formation (*Gazit, 2002*; *Mossou et al., 2014*; *German, Uyaver & Hansmann, 2015*; *Do, Kincannon & Bowers, 2015*). π-stacking has been reported to create optically active aggregates (*Kuo, 2011*; *Handelman et al., 2016*) and is responsible for large red-shifts of fluorescence emissions through the formation of J-aggregates (*McRae & Kasha, 1958*; *Higgins & Barbara, 1995*; *Uemori et al., 2012*; *Cao et al., 2014*), which could overlap with amyloid-bound dye fluorescence. In this work we generated Phe aggregates using previously described methods and analyzed their spectroscopic profiles in order to determine whether they interacted with ThT or formed π-stacking optically active structures.

## MATERIALS AND METHODS

### Phenylalanine sample preparation

A total of 100 nM Phe samples were prepared by dissolving Phe in 50 mM (pH 6) phosphate buffer at 25 °C. Phe aggregation was carried out at 60 °C with no sample agitation for 15 days.

### Fluorescence measurements

Solution fluorescence spectra were recorded using an excitation range of 200–500 nm, emission range of 250–650 nm and wavelength increment of five nm, with excitation slit—five nm, emission slit—five nm for incubation observations and excitation slit—10 nm, emission slit—five nm for final aggregate solution analysis with a Varian Cary Eclipse spectrophotometer. Each spectrum was averaged from three separate measurements with subtracted 50 mM (pH 6) phosphate buffer spectra.

### Aggregate filtration and sonication

Regenerated cellulose filters (0.22 μm) and Amicon concentrators (3 kDa) were used to remove different size of aggregates. Sonication of aggregates was performed using Bandelin Sonopuls 3100 sonicator equipped with a MS72 tip for 20 s at 20% intensity.

### Hydrogel preparation

Hydrogels were prepared by heating a 50 mM (pH 6) phosphate buffer to 100 °C and dissolving agarose to a concentration of 1% or Phe to a concentration of 300 mM and adding ThT to a concentration of 50 μM. ThT fluorescence was measured before (at 80 °C) and after (at 25 °C) hydrogel formation using 440 nm wavelength excitation and

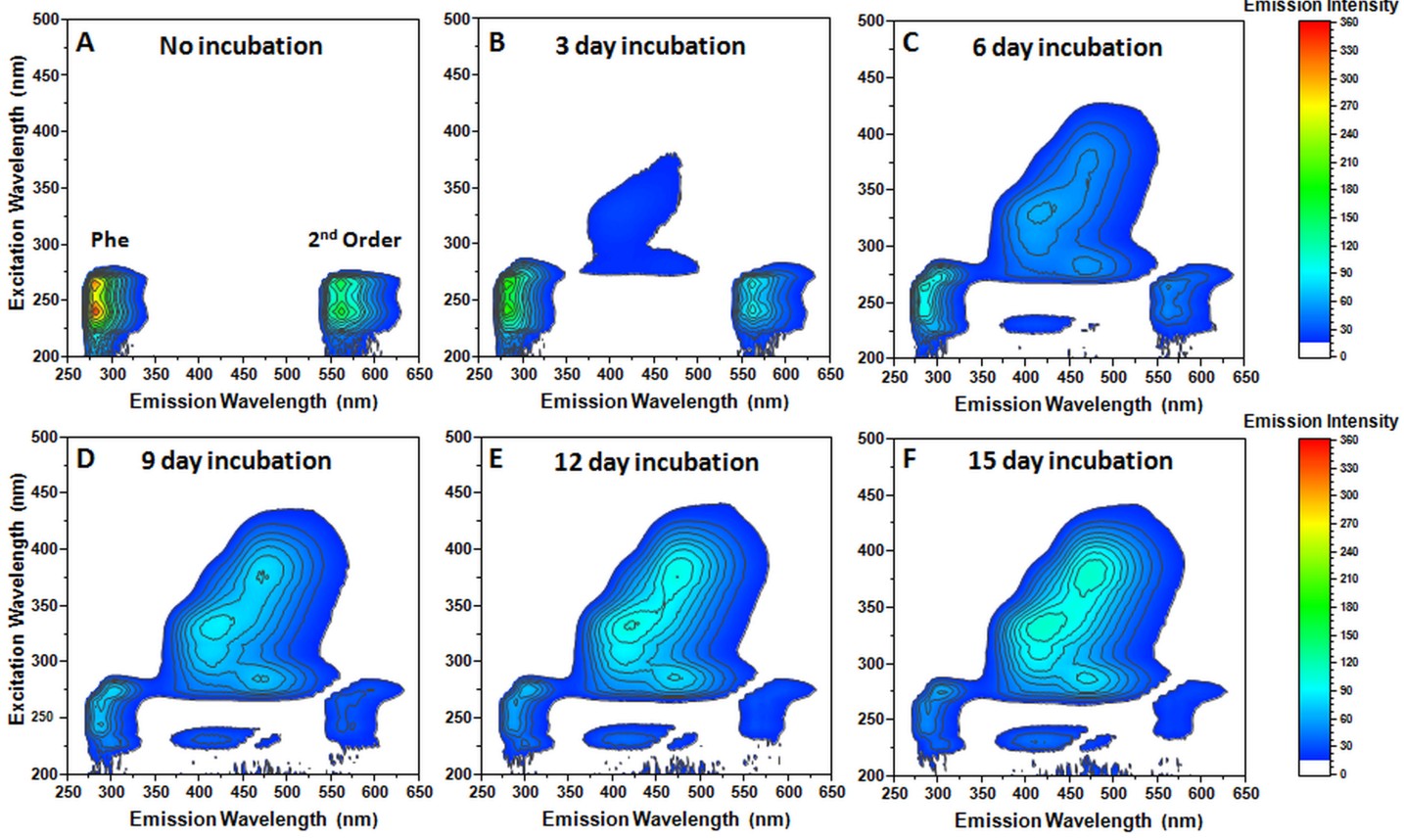

**Figure 1 Excitation/emission spectra of 100 mM L-phenylalanine measured every 3 days during a 15-day incubation period.** Excitation/emission spectra without incubation (A), with 3 day (B), 6 day (C), 9 day (D), 12 day (E) and 15 day (F) incubation. The examined spectra region contains a second order diffraction artefact of Phe (marked in the no incubation spectra). Five nm excitation and emission slits were used.

470–510 nm emission range with excitation slit—10 nm, emission slit—5 nm. Each spectrum was averaged from three separate measurements.

### Thioflavin T sample spectra measurements

One µL of 10 mM ThT solution was added to 199 µL of 100 mM Phe aggregate solution, to a final ThT concentration of 50 µM. The 3D spectra were measured using the previously described method.

## RESULTS AND DISCUSSION

A total of 100 mM Phe samples were incubated at 60 °C and their excitation/emission spectra were measured every 3 days (Fig. 1). The initial samples with no incubation only contained Phe-specific emissions (Fig. 1A). After 3 days of incubation, a large region of low-intensity fluorescence emission appeared on both near-UV and visible light spectrum (Fig. 1B). Such emissions were also seen by *Tikhonova et al. (2018)* at lower Phe concentrations and attributed to deep-blue autofluorescence from amino acid oxidation. Continuous incubation leads to an increasing intensity of these emissions and further expansion into longer wavelength regions (Figs. 1C–1F). There was also a
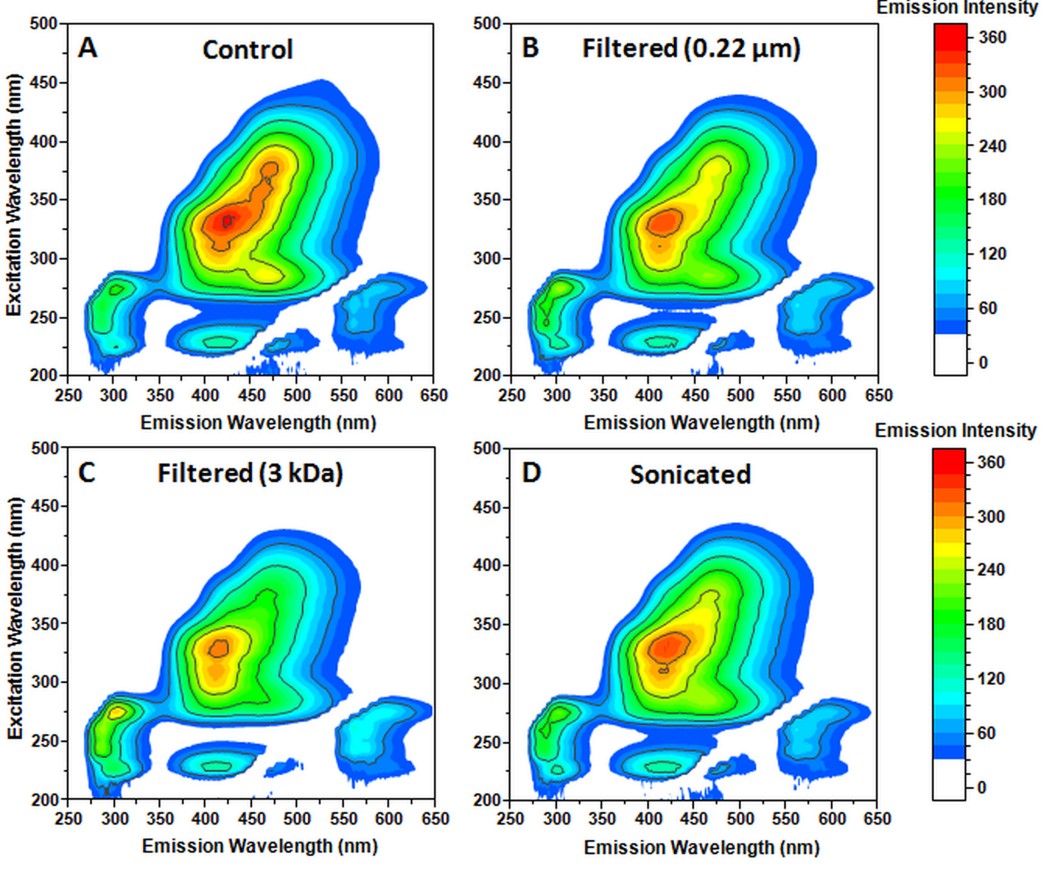

**Figure 2 Effect of filtration and sonication on the spectra of aggregated Phe.** Excitation/emission spectra of the Phe aggregate sample: control (A), after filtration through a 0.22 µm pore size filter (B) or a 3 kDa concentrator (C) and after sonication (D). Ten nm excitation and five nm emission slits were used.

noticeable decrease in Phe-specific fluorescence emission intensities. After 12 days of incubation, the spectra no longer experienced any substantial changes upon further incubation. The newly formed fluorescence excitation/emission region had multiple maxima positions (375/472, 360/461, 285/467, 230/418, 330/425 and 310/419 nm).

Dynamic light scattering examination of the final solution revealed the appearance of large particles (Supplementary Fig. 1). There was also a substantial increase in the sample's optical density over the examined wavelength region (Supplementary Fig. 2). In a paper by *Do, Kincannon & Bowers (2015)* a theoretical model of the aggregates composed of Phe tetramer units was proposed. These units could interact through π-stacking, creating a fibrillar structure. The newly emerged emission bands would be the result of exciton exchange between aromatic rings. The appearance of longer wavelength emissions following incubation could be the result of J-aggregate-type interactions between these tetramer units, forming larger structures. Surprisingly, the fluorescence region was partially overlapping with the excitation/emission maxima of amyloid-bound ThT (*Voropai et al., 2003*). This could cause false-positive ThT assay results and give an inaccurate impression of amyloid fibril formation, despite it being π-aggregate assembly.

**Table 1 Fluorescence emission intensity changes of Phe-specific, aggregate-specific, and FRET maxima.**

| Excitation | Emission | Control Intensity | Filtered (0.22 µm) Intensity | Change (%) | Filtered (3 kDa) Intensity | Change (%) | Sonicated Intensity | Change (%) |
|---|---|---|---|---|---|---|---|---|
| 275 | 303 | 204 ± 3 | 241 ± 17 | 18 ± 7 | 285 ± 21 | 40 ± 8 | 219 ± 2 | 8 ± 2 |
| 260 | 290 | 180 ± 2 | 202 ± 11 | 13 ± 6 | 240 ± 20 | 33 ± 8 | 197 ± 2 | 9 ± 1 |
| 245 | 288 | 163 ± 4 | 187 ± 11 | 15 ± 6 | 223 ± 20 | 37 ± 9 | 179 ± 2 | 10 ± 3 |
| 225 | 304 | 127 ± 7 | 142 ± 7 | 12 ± 8 | 170 ± 16 | 34 ± 11 | 137 ± 4 | 8 ± 6 |
| 375 | 472 | 312 ± 1 | 266 ± 7 | −15 ± 3 | 187 ± 22 | −40 ± 12 | 248 ± 3 | −21 ± 1 |
| 360 | 461 | 311 ± 1 | 266 ± 11 | −14 ± 4 | 203 ± 20 | −35 ± 10 | 267 ± 2 | −14 ± 1 |
| 330 | 425 | 347 ± 6 | 324 ± 7 | −7 ± 3 | 303 ± 15 | −13 ± 5 | 330 ± 2 | −5 ± 2 |
| 310 | 419 | 310 ± 3 | 297 ± 6 | −4 ± 2 | 294 ± 8 | −5 ± 3 | 304 ± 1 | −2 ± 1 |
| 285 | 467 | 270 ± 3 | 232 ± 6 | −14 ± 3 | 187 ± 13 | −31 ± 6 | 239 ± 1 | −11 ± 1 |
| 230 | 418 | 139 ± 2 | 134 ± 3 | −3 ± 3 | 138 ± 2 | −1 ± 2 | 137 ± 2 | −1 ± 2 |

**Note:**
Filtration- and sonication-induced fluorescence emission intensity changes for Phe-specific (green), aggregate-specific (red) and FRET (blue) maxima.

The appearance of optical properties (Fig. 2A) could be the result of chemical modifications that occur to Phe upon incubation at 60 °C (*Tikhonova et al., 2018*). In order to test such a possibility, we attempted to remove the aggregates using filtration through 0.22 µm filters (Fig. 2B). We also utilized a 3 kDa concentrator to remove any smaller aggregates (Fig. 2C), as there were reports indicating that their size could be as low as trimers (*Spitz et al., 2002*; *Kuo, 2011*). The sample was also sonicated to examine the aggregate assembly stability (Fig. 2D).

Filtration through a 0.22 µm filter resulted in a decrease of the six excitation/emission maxima associated with the newly formed aggregates and an increase in Phe-specific band intensity (Table 1). A similar result was achieved after the sample's sonication, however, there was a noticeably larger decrease in intensity at the 375/472 maxima, which could indicate that some smaller aggregates that are able to pass through a 0.22 µm filter can be disassembled. There was also no apparent increase in any of the aggregate-specific maxima, which could mean that all aggregates are affected by sonication, since any larger structure disassembly would lead to an increased band intensity of the smaller assemblies.

Filtration through a 3 kDa concentrator yielded a much greater decrease in fluorescence of the aggregate-specific bands and a sizeable increase in Phe-specific band intensity. Interestingly, while the increase in Phe-specific band intensities was similar for all four measured maxima positions, there was a disparity between the remaining six maxima intensity changes. The 375/472, 360/461 and 285/467 position intensities decreased significantly, while 230/418, 330/425 and 310/419 position intensities experienced a minimal change. This means that the resulting spectra may be composed of both chemically modified Phe (at positions that experience minimal change) and Phe aggregates. It was clear that larger aggregates had specific excitation/emission characteristics and that there was a correlation between the emission intensity increase of Phe-specific maxima positions and the decrease in aggregate-specific maxima positions.

After an examination of the excitation/emission properties of the Phe-solution spectra we can see that the Phe-specific emission bands overlap with the aggregate-specific excitation

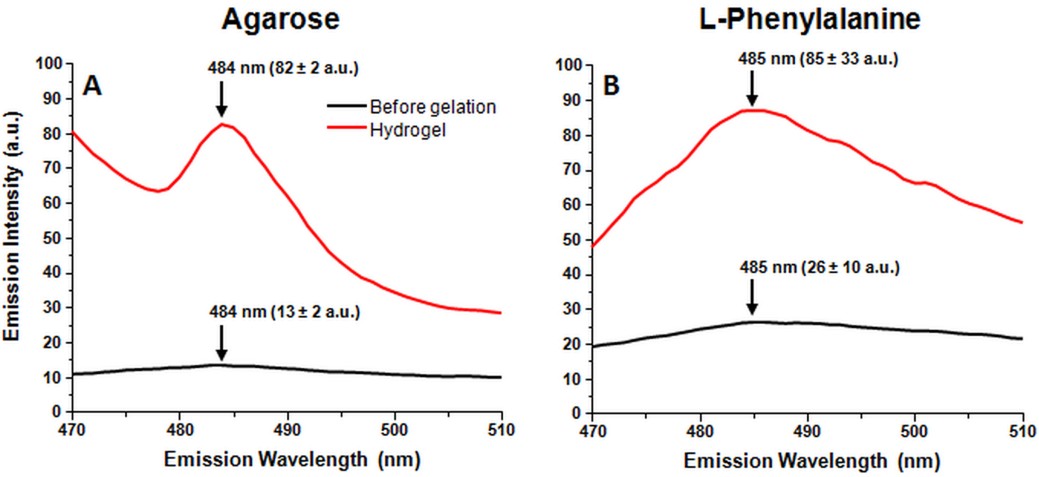

**Figure 3** One percent agarose and 300 mM Phe gel formation induced ThT fluorescence emission changes (sample excitation wavelength – 440 nm). Fluorescence emission intensity changes observed during one percent agarose (A) and 300 mM Phe (B) gel formation. Ten nm excitation and five nm emission slits were used. Emission intensity was calculated from three separate measurements.

wavelengths. Such an overlap could lead to Forster resonance energy transfer (FRET) between non-aggregated Phe molecules, chemically modified Phe and Phe assemblies. Upon a detailed analysis, it seems likely that the Phe-specific emissions can be absorbed by either aggregates or modified Phe and lead to an emission maximum at 230/418. The 230/418 FRET emissions could be caused by Phe molecules and 330/425 and 310/419 bands which are related to smaller assemblies or modified Phe and these FRET emissions experience almost no reduction in intensity upon filtration or sonication. The 285/467 FRET emissions, however, could be related to larger aggregates and diminish greatly upon filtration or sonication. The FRET hypothesis could explain why the removal of aggregates causes an increase of Phe-specific fluorescence.

A hydrogel fluorescence assay was also conducted to check whether the previously reported increase in ThT fluorescence intensity upon Phe hydrogel formation was due to ThT immobilization or amyloid formation. Two hydrogels containing ThT were prepared, one using a 300 mM Phe solution and another with 1% agarose. Both assays were conducted under the same heating and subsequent cooling conditions and their ThT fluorescence emission spectra were measured during the cooling process.

In the case of agarose gel formation (Fig. 3A), a fluorescence emission maximum appeared at 484 nm and an increase in ThT fluorescence at this maximum increased from 13 ± 2 a.u. to 83 ± 2 a.u. (>6-time increase). As for the 300 mM Phe solution (Fig. 3B), the emission maximum was at 485 nm and the increase in ThT fluorescence was from 26 ± 10 a.u. to 85 ± 33 a.u. (>3-time increase).

With both gels we observe a similar emission maxima position as well as a substantial emission intensity growth. This maxima position correlates with the ones often seen in amyloid fibril staining with ThT (*Voropai et al., 2003*); however, it is quite obvious that agarose does not form amyloid fibrils, as it is not even an amino acid, which leads to the conclusion that such emission spectra are the result of ThT immobilization and not

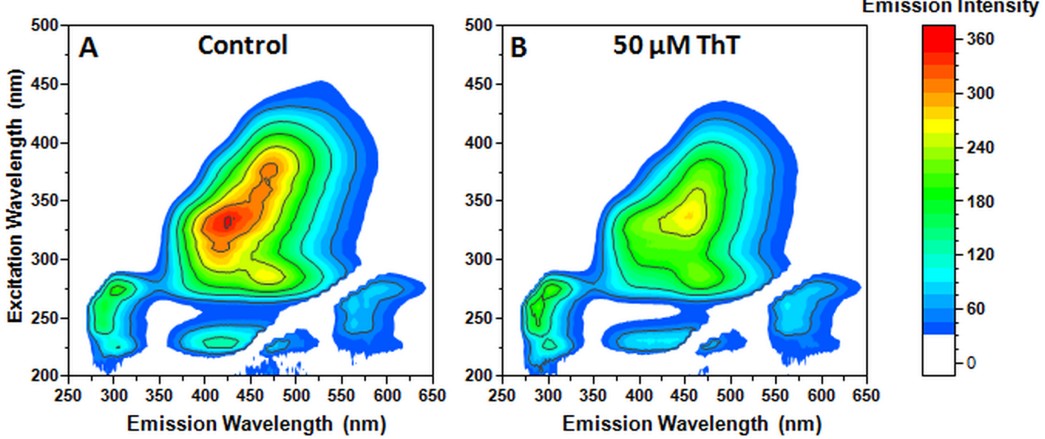

**Figure 4 Aggregated Phe fluorescence emission changes after the addition of 50 µM ThT.** Excitation/emission spectra of aggregated Phe before (A) and after (B) the addition of ThT. Ten nm excitation and five nm emission slits were used.

binding to amyloid fibril surface. Such a coincidence could easily lead to a false-positive result in detecting amyloid fibril formation.

The addition of ThT to the incubated Phe sample (Fig. 4A) resulted in a decrease of Phe aggregate-specific emissions (Fig. 4B). If the solution contained amyloid fibrils, capable of binding ThT, we would see a noticeable increase in fluorescence emissions at the 440/480 excitation/emission region. The observed decrease in the region's fluorescence emission intensity can be explained by the absorption of aggregate-specific emissions by ThT molecules, which in turn do not exhibit fluorescence, due to not being bound to any amyloid fibrils.

These results show that in the case of Phe aggregates generated under these conditions, ThT acts as a fluorescence quenching agent and actually hinders the detection of these structures.

## CONCLUSIONS

These findings show that Phe does indeed form aggregates, as was demonstrated on multiple occasions, however, autofluorescence of the aggregates, described in this work, partially overlaps with the spectra of dyes used in amyloid fibril assays. Furthermore, the formation of hydrogels can lead to immobilization of ThT and result in an increased emission intensity, which is not the result of interactions with beta-sheet structures.

### Funding

The authors received no funding for this work.

### Competing Interests

The authors declare that they have no competing interests.
## Author Contributions

- Mantas Ziaunys conceived and designed the experiments, performed the experiments, analyzed the data, prepared figures and/or tables, authored or reviewed drafts of the paper, approved the final draft.
- Vytautas Smirnovas conceived and designed the experiments, contributed reagents/materials/analysis tools, authored or reviewed drafts of the paper, approved the final draft.

## Data Availability

The raw data are available in the Supplemental Files.

## Supplemental Information

Supplemental information for this article can be found online at http://dx.doi.org/10.7717/peerj.6518#supplemental-information.

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
