# Peer review of "Emergence of visible light optical properties of L-phenylalanine aggregates"

_PeerJ, doi:10.7717/peerj.6518_

## Round 0.1 · original submission · Major Revisions

Please address all the critical points raised by the reviewers and mend your manuscript accordingly.

Reviewer 1 ·

Basic reporting

All the formal criteria are met.

Experimental design

Though testing of the impact of intrinsic fluorescecne to the ThT signal is meaningful, I have several questions about the experimental design, which are described in the General Comments section. The authors have to answer this questions before the paper is published.

Validity of the findings

The main conclusion of the paper is that ThT fluorescence can be enhanced due to gelation, but not to fibrillation of Phe. I agree with this fact, as well as with the potential overlap between dbAF and ThT.

Additional comments

1. The authors test the oxidation hypothesis suggested by Tikhonova et al. by using filtration of the samples. However, Tikhonova et al. used mush lower concentrations of Phe to minimize aggregation of the sample. Hence, this comparison is not totally correct.

2. In Figure 1 authors demonstrate the band in EEM, which is an obvious aftefact (second order of difraction, see the peek centered at ca ex = 250/em = 550 nm). Why?

3. During long incubation of Phe (15 days) even at 60C bacterial contamination is possible, which may lead to fluorescence signal close to dbAF of Phe or its aggregates. Hence, sodium azie is used. Please, explain why you don't consider the possibility of bacterial contamination – the structure of bands in EEM is more typical for bacteria than for dbAF of Phe or proteins.

4. "The appearance of optical properties could be the result of chemical modifications that occur to
Phe upon incubation at 60 °C(Tikhonova et al., 2018). In order to rule out such a possibility, we
attempted to remove the aggregates using filtration through 0.22 μm filters" - as can be seen from Table 1, this possibility was not ruled out, but proved - filtration didn't remove 310/419 nm fluorescence, which doesn't originate from Phe. Please, indicate this in the text in a more clear way.
And, once again, see the comment 1 – much higher concentrations are used in this study compared to Tikhonova et al.

5. The FRET hypothesis is speculative now and coud be tested e.g. by titration of Phe aggregates with Phe. However, as a hypothesis it is ok.

6. Please indicate what is the baseline (solution parameters) in Fig. 3.

7. Conclusion: "These findings show that Phe does indeed form aggregates". Please explain, which data in the paper illustrate this fact. I have no doubts that at 100 mM concentration there will be aggregation, however, it is not clear from Fig. 1-2 or Table 1, where the maximum change is 40% at 3kDa filtration. For instance, this conclusion would be much more clear if you show e.g. DLS or other scattering data.

Reviewer 2 ·

Basic reporting

Review on Manuscript Submitted to PeerJ
Manuscript: 32513-v0

Manuscript Title:
"Emergence of visible light optical properties of L-phenylalanine
aggregates"

Authors:
M. Ziaunys, V. Smirnovas


I have read the paper with a great interest. On the one hand it considers well known experimental biomedical research of formation of amyloid fibrils. On the other hand it teaches the researchers how to avoid false conclusions relying on misleading data based on visible photon emission.
Two kinds of visible luminescence techniques (effects, methods) allowing to detect and monitor nucleation and growth of amyloid structures are considered. The first one is visible autofluorescence created by hydrogen bonds of -sheet secondary structure. The second method applies visible dye imaging labels such as ThT and Congo red which are the major assay to observe Phe amyloid fibril formation.
The authors conducted good and reliable experimental research.
Their studies of the excitation/emission properties of the Phe solution spectra show that the Phe-specific emission bands overlap with the aggregate specific excitation wavelengths. This could cause false-positive ThT assay results and give an inaccurate impression of amyloid fibril formation, despite it being π-aggregate assembly. The results also directly indicate that in the case of Phe aggregates, ThT could acts as a fluorescence quenching agent and actually hinders the detection of these structures.

My opinion: It is a good methological work. It can be accepted and published as is.

Experimental design

no comment

Validity of the findings

no comment

Reviewer 3 ·

Basic reporting

The manuscript is well written. Literature references and background information are fine.

Experimental design

Overall the design is solid and the methods are fine. One question I have is the difference in fluorescence intensities between Fig. 1 and Fig. 2. Were the Phe concentrations or other conditions different? It is not clear from the manuscript.

Validity of the findings

The results are fine but I think the authors may be overstating their case somewhat. It is clear from the manuscript that Phe forms some kind of aggregates in the conditions used and these aggregates are auto fluorescent. But it's not clear what is the relationship between these aggregates and any previously reported ThT-positive aggregates of Phe. As it is well known, depending on the conditions proteins and peptide can form a variety of aggregates and only relatively few of them bind ThT effectively. I think the authors could clarify this.

---

## Round 0.2 · Minor Revisions

Please address the remaining issues pointed by the reviewer #1 and amend your manuscript accordingly.

Reviewer 1 ·

Basic reporting

The authors have addressed my major concerns in the revised manuscript. Some additional remarks can be found in the "General comments for the author" section.

Experimental design

no comment

Validity of the findings

no comment

Additional comments

1. "Normally we would consider not including the region beyond 2nd order Rayleigh scattering as it is almost always filled with 2nd order artefacts, however, the 230/418 band (which is likely a FRET band) extends slightly into the region beyond 2nd order Rayleigh scattering and its extent into this region varies after the reduction of the newly appeared bands (see included Fig. R1), hence we feel like it should be included. Which means the artefacts that happen to be in the scanned region also get included."

This could be due only to higher scattering in the "control" sample compared to the "filtered" one, you can check this by comparing the intensities at 460 nm (230/460).

2. "Nevertheless, we prepared Phe samples with 0.1% sodium azide. As expected, the spectra were slightly affected by sodium azide, but non-phenylalanine bands were still present (Fig. R3) as well as was the overlap with ThT specific emissions at 440/480 nm."

The authors did not provide the incubation time and the conditions of the experiment. Their results thus can not be critically analyzed. The EEM in Fig. R3 is definitely different from Fig. 1 in the manuscript and this has to be commented.

Also I strongly suggest to the authors to include the optical density spectra for the incubated Phe samples to show the inner filter effect influence on the presented data – it seems that it would be quite high.

3. Conclusions
"These findings show that Phe does indeed form aggregates, as was demonstrated on multiple
occasions, however, aggregates generated under the conditions used in this work have optical
properties that give rise to a false identification as amyloids when applying ThT assays."

Better correct to "false identification of Phe-based fibrils", as the general statement about amyloids is out of scope of this paper.

Reviewer 3 ·

Basic reporting

No comment

Experimental design

No comment

Validity of the findings

No comment

Additional comments

Overall, looks good.

---

## Round 0.3 · accepted · Accept

All the critical issues were adequately addressed and the manuscript is acceptable now.

#